# An Improved Passive CR-39-Based Direct ^222^Rn/^220^Rn Progeny Detector

**DOI:** 10.3390/ijerph17228569

**Published:** 2020-11-18

**Authors:** Jun Hu, Guosheng Yang, Chutima Kranrod, Kazuki Iwaoka, Masahiro Hosoda, Shinji Tokonami

**Affiliations:** 1Institute of Radiation Emergency Medicine, Hirosaki University, 66-1 Hon-cho, Hirosaki 036-8564, Japan; hujun@hirosaki-u.ac.jp (J.H.); kranrodc@hirosaki-u.ac.jp (C.K.); m_hosoda@hirosaki-u.ac.jp (M.H.); 2National Institutes for Quantum and Radiological Science and Technology, 4-9-1 Anagawa, Inage, Chiba 263-8555, Japan; yang.guosheng@qst.go.jp (G.Y.); iwaoka.kazuki@qst.go.jp (K.I.); 3Graduate School of Health Sciences, Hirosaki University, 66-1 Hon-cho, Hirosaki 036-8564, Japan

**Keywords:** ^222^Rn progeny, ^220^Rn progeny, CR-39, equilibrium equivalent concentration, deposition velocity

## Abstract

An improved passive CR-39-based direct ^222^Rn/^220^Rn progeny detector with 3 detection channels was designed and tested in this study to measure and calculate equilibrium equivalent concentration (EEC) of both ^222^Rn and ^220^Rn without the equilibrium factor. A theoretical model was established to calculate the EEC with optimization. Subsequently, an exposure experiment was carried out to test the performance of this detector, and we compared the chamber experiment and the theoretical model by estimating and measuring various parameters. The deposition flux of progeny derived from the prediction agreed well with the value measured in the exposure chamber. The energy-weighted net track density (NTD) measured by this detector is much more reliable to reflect the linear relation between NTD and time-integrated EEC. Since the detector is sensitive to the exposure environmental condition, it is recommended to apply the detector to measure the EEC after its calibration in a typical indoor environment.

## 1. Introduction

In the conventional integrating measurements for ^222^Rn and ^220^Rn activity concentrations in a large-scale survey, the solid-state nuclear track detector (SSNTD) technique is usually used. Several designs of measuring devices using SSNTD have been used for indoor ^222^Rn and ^220^Rn surveys, such as RADUET [1], Pin-hole dosimeter [2,3,4], and so on. In the two different diffusion rate chambers of these measuring devices, because there is a diffusion barrier against ^220^Rn based on its quite short half-life, the low diffusion rate chamber detects mainly ^222^Rn and the high diffusion rate chamber detects both ^222^Rn and ^220^Rn. However, evaluation of the internal exposure to ^222^Rn and ^220^Rn does not directly use the activity concentration of ^222^Rn and ^220^Rn gas instead of the equilibrium equivalent concentration (EEC). It should be noted that the equilibrium factor depends on various parameters, such as the radioactive decay, ventilation, and reactions with the structure and the surface of furnishing. Because of the short half-life of ^220^Rn (55.6 s), the indoor ^220^Rn concentration distributes heterogeneously, decreasing with distance from the source [5,6]. Therefore, it is not feasible to use the ^220^Rn concentration directly measured by the integrating detectors to evaluate its internal exposure. Otherwise, study of the behavior of ^222^Rn and ^220^Rn decay products indoors is important for assessing the natural background radiation exposures on the public through the inhalation route. Therefore, the direct measurement technique of ^222^Rn and ^220^Rn progeny is desirable and necessary for the evaluation of internal indoor ^222^Rn and ^220^Rn exposure.

In the recent 20 years, a direct ^220^Rn progeny measurement technique based on CR-39 detector for EEC of ^220^Rn was developed and used by Zhuo and Iida [7], Tokonami et al. [8], Zhuo and Tokonami [9], Tokonami [10], Sorimach et al. [11], and Hu et al. [12]. Mishra and Mayya [13] were the first to develop the direct ^222^Rn and ^220^Rn progeny measurement techniques using LR-115 detector. CR-39 is the most sensitive and popular detector for recording α-particles [14] and performs stable results; no direct ^222^Rn and ^220^Rn progeny detector using CR-39 was developed and applied because its energy windows for recording α-particles is quite wide, and it is unable to distinguish alpha energy directly, although some researchers make great efforts to develop dedicated tools based on the track geometrical characteristics to measure alpha particles. Moreover, the application of the direct measurement technique for determining their progeny concentrations depends on estimation of the effective deposition velocities of combinations of ^222^Rn and ^220^Rn progenies in a typical indoor environment. Some researchers [11,15] used the experimental method to estimate the geometric mean deposition velocity and applied the data to actual measurement in an indoor environment. However, the experimental conditions had a lot of limitations on environmental parameters for estimating the deposition velocity. In this study, we developed a passive CR-39-based direct ^222^Rn/^220^Rn progeny detector. The ^222^Rn and ^220^Rn behavior model and particle deposition model were used to estimate the concentration of each ^222^Rn and ^220^Rn progeny as well as the EEC.

## 2. Materials and Methods

### 2.1. The Passive CR-39-Based Direct ^222^Rn/^220^Rn Progeny Detector

The structure of the detector is shown in Figure 1. The piece of commercially available ally diglycol carbonate (CR-39) (the BARYOTRAK, produced by Fukuvi Chemical Industry Co., Ltd., Fukui, Japan) was used as a detecting material to measure alpha particles emitted from ^222^Rn and ^220^Rn progeny. Each detection channel was mounted with a different thickness of aluminum-vaporized polyethylene films (Mylar films), which can only detect the alpha particles emitted from corresponding progeny radionuclides and ensure that lower energy alpha emissions (from the gases and other airborne alpha emitters) do not pass through the absorber. In the conventional passive direct ^222^Rn/^220^Rn progeny detector, there are only 2 detecting channels: the ^220^Rn progeny (TnP) channel and the ^222^Rn progeny (RnPII) channel. The TnP channel can only detect the 8.78 MeV alpha particles emitted from ^212^Po atoms, which are formed from the decay of ^212^Pb and ^212^Bi atoms deposited on the film surface without any interference from other alpha emissions of natural radioactivity decay series. The RnPII channel selectively detects ^212^Po and ^214^Po (7.69 MeV) from the decay of surface deposited ^222^Rn and ^220^Rn progeny. In the improved passive detector, we added one more detecting channel RnPI to detect the alpha emission energy higher than that of ^218^Po (6.0 MeV). The design information is present in Table 1. Since the RnPI and RnPII cthannels are in a mixed ^222^Rn and ^220^Rn progeny environment, there are interferences from the high energy alpha particles, which could be subtracted using the tracks on the TnP channel. Therefore, 4 slices of CR-39 detector are used on the TnP channel instead of 2 slices on the other channels to decrease the error.

### 2.2. The Theoretical Model and Parameters

#### 2.2.1. Theoretical Model

Among the several aspects of indoor behaviors of ^222^Rn and ^220^Rn, apart from the decay losses, deposition and ventilation are two major mechanisms to remove the decay products from indoor air. Unlike the removal process by ventilation which depends essentially on the air exchange rate in a given room environment, the removal by deposition depends on the activity size distribution and the structure of turbulence at the air–surface interface. Since this detector is designed using the deposition mode, it is necessary to avoid uncontrolled static charges from affecting the deposition rates. Hence, the aluminized side of the Mylar is chosen to act as the deposition surface. 

To describe the behavior of ^222^Rn and ^220^Rn progeny, the relationship between the attached ^222^Rn and ^220^Rn progeny deposition flux, *J* (atom cm^−2^ s^−1^), and track density, *N* (tracks cm^−2^) can be expressed as follows:(1)$$J=\frac{N}{\eta t}$$
where *η* is the track registration efficiency, which is the multiplier of the branching ratio of its progeny, the geometric efficiency, and etching efficiency. For each channel, the geometric efficiency depends on the energies of incident α-particle, incident angles against the absorbers, and the thickness of the absorbers. The track registration efficiency measured in this study is present in Table 1, which corresponds to the etching condition using 6 N NaOH solutions at 60 °C for 24 h without stirring. *t* is the exposure period (s).

The effective deposition velocity of the progeny, *V_e_* (cm s^−1^), is defined as follows:(2)$$V_{e}=\frac{J}{C}$$
where *C* is the atom concentration of the progeny (atom cm^−3^).

To determine the deposited progeny atoms from the alpha tracks registered in CR-39, we need to check the progeny atoms’ ultimate decay on the surface of the detector, which causes the alpha track to the CR-39. In the case of ^222^Rn progeny, a fraction of all the alpha particles emitted from the atoms of ^218^Po, ^214^Pb, and ^214^Bi deposited on the absorber can form tracks on the CR-39. Therefore, the tracks counted at the end can be directly proportional to the sum of the ^218^Po, ^214^Pb, and ^214^Bi atoms deposited. Similarly, the total tracks registered by the ^220^Rn progeny will be proportional to the sum of the ^212^Pb and ^212^Bi atoms deposited. Accordingly, we can define the following:(3)$$N_{Tn}=\eta_{Tn,Po212}\cdot V_{e,Tn}\cdot \left(C_{Pb212}+C_{Bi212}\right)\cdot t$$
(4)$$N_{Rn2}=\eta_{Rn2,Po212}\cdot V_{e,Tn}\cdot \left(C_{Pb212}+C_{Bi212}\right)\cdot t+\eta_{Rn2,Po214}\cdot V_{e,Rn}\cdot \left(C_{Po218}+C_{Pb214}+C_{Bi214}\right)\cdot t$$
(5)$$\begin{array}{c} N_{Rn1}=\eta_{Rn1,Po212}\cdot V_{e,Tn}\cdot \left(C_{Pb212}+C_{Bi212}\right)\cdot t+\eta_{Rn1,Po214}\cdot V_{e,Rn}\cdot \left(C_{Po218}+C_{Pb214}+C_{Bi214}\right)\cdot t\\ \;+\eta_{Rn1,Bi212}\cdot V_{e,Tn}\cdot \left(C_{Pb212}+C_{Bi212}\right)\cdot t+\eta_{Rn1,Po218}\cdot V_{e,Rn,Po218}\cdot C_{Po218}\cdot t \end{array}$$
where the subscripts of *Tn*, *Rn1*, and *Rn2* mean the parameters of the TnP channel, RnPI channel, and RnPII channel, respectively, and the subscript of each radionuclide means the corresponding parameter. Some of the parameters used in this study are shown in Table 1. *V_e,Tn_* is the effective deposition velocity of ^220^Rn progeny, which is a combination of deposition velocities of ^212^Pb and ^212^Bi. *V_e,Rn_* is a combination of deposition velocities of ^218^Po, ^214^Pb, and ^214^Bi. *V_e,Rn,Po218_* is the only effective deposition velocity of ^218^Po.

In the mixed ^222^Rn and ^220^Rn environment, we can easily count the registered tracks on the TnP channel from ^212^Po decay, which come from the decay of the deposition flux of ^212^Pb and ^212^Bi, as shown in Equation (6). Accordingly, from Equations (3)–(5), we can also calculate the counts of registered tracks from ^214^Po on the RnPII channel and ^218^Po on the RnPI channel as in Equations (7) and (8).
(6)$$N_{Po212}^{TnP}=\eta_{Tn,Po212}\cdot V_{e,Tn}\cdot \left(C_{Pb212}+C_{Bi212}\right)\cdot t=N_{Tn\;}$$
(7)$$N_{Po214}^{RnP2}=\eta_{Rn2,Po214}\cdot V_{e,Rn}\cdot \left(C_{Po218}+C_{Pb214}+C_{Bi214}\right)\cdot t\;=N_{Rn2}-\frac{\eta_{Rn2,Po212}}{\eta_{Tn,Po212}}N_{Tn}$$
(8)$$N_{Po218}^{RnP1}=\eta_{Rn1,Po218}\cdot V_{e,Rn,Po218}\cdot C_{Po218}\cdot t\;=N_{Rn1}-\frac{\left(\eta_{Rn1,Po212}+\eta_{Rn1,Bi212}\right)}{\eta_{Tn,Po212}}N_{Tn}-\frac{\eta_{Rn1,Po214}}{\eta_{Rn2,Po214}}N_{Po214}^{RnP2}$$

In the atmosphere, a large fraction of ^218^Po and ^212^Pb can react with the ions in the air and form clusters in a short time by the neutralization process. Due to the larger diffusion coefficient of these cluster particles, ^222^Rn and ^220^Rn progenies can attach themselves to various surfaces, such as the surfaces of aerosol particles and droplets in the atmosphere, thereby giving rise to a consecutive activity size distribution. This distribution is broadly classified into two groups: the unattached fraction and the attached fraction. The effective deposition velocities combine the contribution from both the unattached and attached fractions of each progeny species. Considering the contributions of these two fractions, the deposition velocity of each radionuclide can be demonstrated as follows:(9)$$V_{i}=p_{i}\cdot V_{d}^{u}+\left(1-p_{i}\right)\cdot V_{d}^{a}$$
where *p_i_* denotes the unattached fraction of radionuclide *i*, and $V_{d}^{u}$ and $V_{d}^{a}$ denote the deposition velocities of the unattached and attached fraction of airborne particles, respectively, which are dependent on the aerodynamic factors instead of the radionuclides themselves.

Since the registered tracks from ^212^Po decay come from the decay of the deposition flux of ^212^Pb and ^212^Bi, according to the concept of the effective deposition velocity, the deposition velocity of TnP channel could be written as follows:(10)$$V_{e,Tn}=\frac{V_{Pb212}\cdot C_{Pb212}+V_{Bi212}\cdot C_{Bi212}}{C_{Pb212}+C_{Bi212}}$$

Similarly, the deposition velocity of the RnPII and RnPI channels could be written as follows:(11)$$V_{e,Rn}=\frac{V_{Po218}\cdot C_{Po218}+V_{Pb214}\cdot C_{Pb214}+V_{Bi214}\cdot C_{Bi214}}{C_{Po218}+C_{Pb214}+C_{Bi214}}$$
(12)$$V_{e,Rn,Po218}=V_{Po218}$$

The atom concentrations in these equations could also be reexpressed in terms of the activity concentrations (*A_i_*, Bq m^−3^) by using the decay constant *λ_i_*, s^−1^. Subsequently, Equations (6)–(8) could be modified as follows:(13)$$N_{Po212}^{TnP}=\eta_{Tn,Po212}\cdot \frac{V_{Pb212}\cdot \lambda_{Bi212}+V_{Bi212}\cdot \lambda_{Pb212}\cdot r_{3}}{\lambda_{Bi212}+\lambda_{Pb212}}\cdot \left(\frac{1}{\lambda_{Pb212}}+\frac{r_{3}}{\lambda_{Bi212}}\right)\cdot A_{Pb212}\cdot t$$
where *r*_3_ = *A*_*Bi*212_/*A*_*Pb*212_.
(14)$$\begin{array}{c} N_{Po214}^{RnP2}=\eta_{Rn2,Po214}\cdot \frac{V_{Po218}\cdot \lambda_{Pb214}\cdot \lambda_{Bi214}+V_{Pb214}\cdot \lambda_{Po218}\cdot \lambda_{Bi214}\cdot r_{1}+V_{Bi214}\cdot \lambda_{Po218}\cdot \lambda_{Pb214}\cdot r_{2}}{\lambda_{Pb214}\cdot \lambda_{Bi214}+\lambda_{Po218}\cdot \lambda_{Bi214}\cdot r_{1}+\lambda_{Po218}\cdot \lambda_{Pb214}\cdot r_{2}}\\ \cdot \left(\frac{1}{\lambda_{Po218}}+\frac{r_{1}}{\lambda_{Pb214}}+\frac{r_{2}}{\lambda_{Bi214}}\right)\cdot A_{Po218}\cdot t \end{array}$$
where *r*_1_ = *A*_*Pb*214_/*A*_*Po*218_ and *r*_2_ = *A*_*Bi*214_/*A*_*Po*218_.
(15)$$N_{Po218}^{RnP1}=\eta_{Rn1,Po218}\cdot V_{Po218}\cdot \frac{A_{Po218}}{\lambda_{Po218}}\cdot t$$

To solve these equations above, we introduced the Jacobi room model [16] for both ^222^Rn and ^220^Rn. Because the steady-state Jacobi room model is a system of linear equations, the number of unknowns, such as the unattached fractions of each progeny radionuclide and the ratios of the progeny concentration, is less than the number of equations. The solutions of the unknowns could be deduced by the parameters in the Jacobi room model, which are the attachment rate constant, *λ_a_* (s^−1^); the rate constant for deposition of unattached progeny, *λ*_d_^u^ (s^−1^); the rate constant for deposition of attached progeny, *λ*_d_^a^ (s^−1^); and the ventilation rate constant, *λ*_v_ (s^−1^). The details of the equations of the Jacobi room model for both ^222^Rn and ^220^Rn behaviors could be found elsewhere [6]. Finally, the *A*_*Pb*212_ can be solved by the data of the TnP channel and *A*_*Po*218_ can be solved by both the RnPI channel and RnPII channel in the equations.

#### 2.2.2. Parameters Estimation

##### Deposition Velocity

In the theoretical model, Lai and Nazaroff’s [17] three-layer model was adopted to simulate the particle deposition. Compared to the other existing formulations of such models based on the pioneering work of Corner and Pendlebury [18], lacking a thorough physical foundation, the proposed model involves three mechanisms of particle transport: Brownian diffusion, turbulent diffusion, and gravitational settling. It predicts deposition to smooth surfaces as a function of particle size and density. Therefore, the only required input parameters are enclosure geometry and friction velocity. This deposition velocity estimation method has already been applied for deposition particles in a room and experimentally verified by Mishra et al. [15].

##### Parameters in the Jacobi Room Model

In practice, the indoor rooms can be regarded as a rectangular cavity. The rate constant for deposition in the Jacobi room model is estimated by using the deposition velocity calculated by Lai and Nazaroff’s three-layer model [17]. For the rectangular cavity, the rate constant for deposition can be written as follows:(16)$$\lambda_{d}^{i}=n_{c}\cdot \frac{v_{d,u}^{i}S_{u}+v_{d,v}^{i}S_{v}+v_{d,d}^{i}S_{d}}{V}$$
where the superscript *i*, with the allowed values *a* and *u*, allows us to distinguish the two important states of ^222^Rn and ^220^Rn progeny: attached and unattached. *S_u_*, *S_v_*, and *S_d_* are the areas of upward-facing surfaces, vertical surfaces, and downward-facing surfaces, respectively; and *n_c_* is the correction coefficient of the surface area [19], which can be used to correct the changes of ratio caused by the increased furnishing surface area in practical application. The deposition velocity of the unattached fraction was interpolated by the result of the numerical integration for fine particles in the reference [17].

The rate constant for attachment in the Jacobi room model reflects the attachment velocity of the unattached ^222^Rn and ^220^Rn progeny to the ambient aerosol. The charging process depends on the electric charge distribution of an aerosol in steady state. In most used theories, attachment is diffusion-controlled as a result of electrostatic attraction and follows the gas kinetic laws [20,21,22]. The expression of the rate constant for attachment can be written as follows:(17)$$\lambda_{a}=\bar{\beta }\cdot N_{o}$$
(18)$$\bar{\beta }=\int_{0}^{\infty }\beta \left(d_{p}\right)Z\left(d_{p}\right)dd_{p}$$
where *N_o_* is the aerosol concentration, reflecting the concentration of condensation nuclei and where *Z*(*d_p_*) is the number size distribution of aerosol for unit aerosols concentration and can be described as a frequency function of the lognormal distribution with the count median diameter (CMD) and geometric standard deviation (GSD, *σ_g_*). *β*(*d_p_*) is the attachment coefficient. The rate constant for attachment was estimated by the compound trapezoid formula with the Latin-hypercube sampling method.

## 3. Results and Discussion

### 3.1. The Comparison of Chamber Test with a Theoretical Model

To verify the feasibility of measuring the EEC, a verification experiment was conducted in ^222^Rn and ^220^Rn exposure chambers located in Hirosaki University, Japan. The chamber verification system includes 4 function cells, as shown in Figure 2. (1) The source generation cell is a ^222^Rn gas and ^220^Rn gas generation system, which employs the natural uranium rock and commercially available lantern mantles as the ^222^Rn and ^220^Rn sources, respectively, to generate and import ^222^Rn and ^220^Rn to the mix chamber. The humidifier in this cell is responsible for maintaining the stability of the inlet gas concentration. (2) The aerosol generation cell contains 3 components, a compressor, a constant output atomizer (TSI, Inc., Aerosol Generator Model 3076), and a diffusion dryer filled with silica gel. The 5000 μg mL^−1^ NaCl solution is employed to generate droplets with the median diameter of 0.3 μm (GSD < 2.0) to mix with ^222^Rn gas or ^220^Rn gas in the mix chamber. (3) The exposure cell includes two chambers: a mix chamber and an exposure chamber. The volumes of the mix chamber and exposure chamber are 150 L and 3.24 m^3^ (dimensions of 2.25 × 1.2 × 1.2 m^3^, L × W × H), respectively. The mix chamber provides a buffer to create more chances for the aerosol particles and gas to form a stable progeny flux to the exposure chamber. To maintain stable concentration in the exposure chamber, the experiment adopted a circulating connection that applies the outlet of the exposure chamber as the inlet of the source generation cell with an airflow rate of 3 L min^−1^. (4) The measurement cell is responsible for measuring progeny concentrations and environmental parameters in the verification experiment. The portable Si-photodiode detector with a sampling flow rate of 0.5 L min^−1^ and an EMD Millipore MF-Millipore AAWG02500 Mixed Cellulose Ester Filter Membrane were used to measure the EEC of ^222^Rn in real-time. The details of the Si-photodiode detector can be found elsewhere [23,24]. The periodic grab sampling method was carried out to replace the portable Si-photodiode detector to measure the EEC of ^220^Rn. The Model 3034 SMPS^TM^ Scanning Mobility Particle Sizer produced by TSI^®^ was adopted to measure the concentration of condensation nuclei, CMD, and its GSD (σ_g_). The concentration of condensation nuclei in the exposure chamber ranged from 2.05 × 10^4^ to 4.32 × 10^4^ particles per cm^3^. The CMD of the aerosol in the exposure chamber ranged from 0.187 to 0.218 μm. The friction velocity was calculated using a fan-turbulence model [11,25], as follows:(19)$$u^{*}=\frac{N_{s}d_{t}^{2}}{V^{1/3}}$$
where *N_s_* is the rotation speed of the fan (2300 rotation min^−1^), *d_t_* is the blade length for rotation (5 cm), and *V* is the volume of the exposure chamber. In this exposure chamber, we set two fans in opposite directions; as a result, we obtained *u** as 23.53 cm s^−1^. In the verification experiment, each channel had 4 slices of CR-39 in the vertical and upward orientations as one group. For the total verification experiment, each exposure period had 3 groups of progeny detectors in the front, middle, and back of the bottom of the exposure chamber. In the theoretical model, the genetic algorithm is adopted to optimize the parameters in the calculation process. The measured values of the concentration of nuclei condensation number, CMD and its GSD, estimated friction velocity, and ventilation rate are the input parameters. Since the friction velocity, ventilation rate, and particle distribution fluctuate and change all the time, the equilibrium value of each parameter during the exposure period should be optimized depending on the different circumstances for each survey. 

A genetic algorithm with an improved goodness-of-fit objective function was adopted to optimize the results of the theoretical method [26]. The improved genetic algorithm was used to maximize the goodness-of-fit objective function *f*(*x*) as follows:(20)$$f\left(x\right)=\frac{\sum_{i=1}^{n}\omega_{i}}{\sum_{i=1}^{n}\frac{\left|x_{1,i}-x_{2,i}\right|}{\bar{x_{i}}}\cdot \omega_{i}}$$
where *n* is the number of objects of the model; *x_1,i,_* and *x_2,i_* are the measured data from actual measurement of EEC; and *ω_i_* is the weight. The measured ranges of the concentration of nuclei condensation number, CMD, and its GSD are the optimized parameters, and the goodness-of-fit function is carried out to decrease the error of the average EEC of ^222^Rn calculated by the calibrated portable Si-photodiode detector and the CR-39-based progeny detector. The average track density of detectors deployed in the upward and vertical orientations was used to calculate the EEC of ^222^Rn by the progeny detector, which is in the same orientation as that placed in use. The same verification experiment was also conducted for ^220^Rn progeny except for the measurement method for the EEC of ^220^Rn, which used the periodic grab sampling method to replace the portable Si-photodiode detector. The comparison results of the progeny detector and experiment are shown in Table 2 and Table 3. In the calculation, we estimated the deposition velocity of total condensation nuclei in the vertical and upward orientations, which were in the same orientation as the setting in the experiment, and the total deposition velocity and deposition flux of each channel. As shown in Table 2, for the ^220^Rn progeny with an exposure time of 36.9 h, the average experimentally estimated deposition velocities in the upward and vertical orientations were 0.193 ± 0.067 and 0.217 ± 0.045 m h^−1^, respectively. Additionally, the model-simulated deposition velocities were 0.243 and 0.238 m h^−1^, respectively, which were located in the ranges of the experimental data. In the experiment, the deposition velocity in the upward orientation presented insignificant variation with that of the vertical orientation. The exposure time also presented an insignificant influence on the deposition velocity between 25.5 and 36.9 h. For deposition flux, the variation of deposition flux on the TnP channel estimated by the model showed excellent agreement with experimental values. As shown in Table 3, for the ^222^Rn progeny, among distinct exposure times of 30, 70, and 102 h, the calibrated deposition velocities varied narrowly in the upward and vertical orientations, with the ranges of 0.123–0.136 and 0.117–0.131 m h^−1^, respectively. The average deposition fluxes in the theoretical model of ^222^Rn progeny on the RnPII channel and of ^218^Po on the RnPI channel were located in the ranges of the experimental values in most cases. The deposition flux was calculated by the net track density (NTD). The calculation process of NTD itself has uncertainty, especially caused by the sensitivity of the CR-39 detector, read-out of the tracks, counting statistics, and assumed background of the CR-39 detectors [11], except for the exposure environment. Additionally, from Equations (6)–(8), the calculation of the tracks registered by lower alpha-energy progeny should exclude the tracks registered by higher alpha- energy progeny on the channel. Therefore, with a thinner absorber filter, a higher uncertainty of the measured deposition flux is present due to propagations of the uncertainty during the calculation of the NTD.

### 3.2. Comparison of the Estimated Deposition Velocities with Previous Studies

Compared with the deposition velocities of the total ^222^Rn and ^220^Rn progeny of the other researchers [7,11,15,27], as shown in Table 4, these values varied widely from 0.117 to 8.64 m h^−1^ and 0.072 to 19.08 m h^−1^, respectively. To analyze the impact factors of the deposition velocity of ^222^Rn and ^220^Rn progenies, except the deposition velocities measured by Bigu [25], other deposition velocities measured in small-size chambers are much higher than that measured in the test rooms, dwellings, or large-size chambers. This can be explained by the turbulence, caused by the surface roughness, friction velocity, and airflow rate on the surface in a small-size chamber, being much more intense than that in a large room. Similarly, in Bigu’s research, the mixing fan was operated to investigate the effect of strong airflow and turbulence on the deposition velocity [24]. When the fan is on, the surface airflow and turbulence would be more intense, accompanying the increasing deposition velocity of the ^222^Rn and ^220^Rn progenies.

The design of TnP detectors from Zhuo and Iida [7] and Sorimachi et al. [11] are the same as that in the present study, and only the measuring environmental conditions are different. The deposition velocity measured in the small-size chamber [11] is significantly higher than that measured in dwellings [7] and the present study. It should be noted that the deposition velocity measured in this study agreed well with that measured in dwellings. Besides, the deposition velocities of the RnP detector measured by Marsh et al. [15] in the test room and dwelling were also comparable to that of the present study. Therefore, the deposition velocities estimated by both experiments and theoretical models are acceptable in this study. Furthermore, the calibration results from this study are suitable to assess the ^222^Rn and ^220^Rn progeny exposure in test rooms and dwellings.

### 3.3. The Relationship between NTD and Time-Integrated EEC

The relationship between the NTD and the time-integrated EEC of ^222^Rn and ^220^Rn is shown in Figure 3. Based on the concept of the potential alpha energy concentration of ^222^Rn and ^220^Rn progeny, the NTD caused by ^212^Po on the TnP channel can be used to estimate the total potential alpha energy of ^212^Po emitted by the deposited ^212^Pb and ^212^Bi. The detectors were exposed to a total of ^220^Rn progeny concentration of 83 ± 20 Bq m^−3^ during the exposure period from 0.2 to 2.5 d. The time-integrated EEC of ^220^Rn ranged from approximately 16 to 168 Bq m^−3^ d, which corresponded to the time-integrated EEC of ^220^Rn in an exposure period between 1 week to 3 months with the average EEC of ca. 2 Bq m^−3^ per day indoors [5]. The result demonstrated that the values of NTD increased linearly with an increase of time-integrated EEC of ^220^Rn. Accordingly, the NTD caused by ^218^Po on channel RnPI can be used to estimate the atom number of the deposited ^218^Po on the detector firstly decaying to ^214^Pb. Furthermore, the NTD of ^214^Po can be used to calculate the total potential alpha energy of ^214^Po emitted by the deposited ^218^Po, ^214^Pb, and ^214^Bi during the decay. Therefore, in Figure 3a, the energy-weighted NTD was calculated as follows:(21)Energy weighted NTD = εP218o·NTD of P218o+εP214o·NTD of P214o εP218o+ εP214o
where *ε* (i) is the alpha energy of the radionuclide *i* emitted during the decay. *ε*(^218^Po) is 6.0 MeV, and *ε* (^214^Po) is 7.7 MeV. The progeny detectors were exposed to an average EEC of ^222^Rn of 685 ± 6 Bq m^−3^ during the exposure period. The time-integrated EEC of ^222^Rn ranged from approximately 863 to 2839 Bq m^−3^ d, which corresponds to the time-integrated EEC of ^222^Rn in an exposure period from 1 month to 3 months with an average EEC of ^222^Rn of ca. 30 Bq m^−3^ per day. As a result, the values of the three NTDs increased linearly with the increase of the time-integrated EEC of ^222^Rn. Thus, it is possible to estimate the EEC of ^222^Rn indoors during a certain period. To compare the three NTDs as a function of the time-integrated EEC of ^222^Rn, the R^2^ of NTD of ^218^Po, energy-weighted NTD, and NTD of ^214^Pb were successively decreased from 0.9861 to 0.5625. Therefore, compared to the other deposition-based passive ^222^Rn progeny detectors which only measure the NTD contributed by ^214^Po, the detector with two channels is much more reliable. Besides, the progeny detector with 4 slices of CR-39 for the TnP channel designed rather than the same slices of CR-39 for each channel is recommended to decrease the error in calculation.

## 4. Conclusions

In this study, a 3-channel passive CR-39-based direct ^222^Rn/^220^Rn progeny detector was developed and tested to measure and calculate the EEC of both ^222^Rn and ^220^Rn without the equilibrium factor. In the verification chamber experiment, the genetic algorithm was implemented to optimize the estimation result of the theoretical model. The optimized deposition fluxes of ^220^Rn progeny on the TnP channel, the ^222^Rn progeny on the RnPII channel, and ^218^Po on the RnPI channel agreed well with the experimental measurements. The EEC of ^220^Rn and ^222^Rn measured by the calibrated devices are comparable to the estimated value of the detector. Compared with the two channels’ detectors, the energy-weighted NTD is much more reliable to estimate the EEC of ^222^Rn rather than only measuring the NTD contributed by ^214^Po. Moreover, because the exposure conditions, especially the airflow and turbulence in the room, are smaller than that in the small-size chamber, the deposition velocity measured in this study is close to the value measured in the room. Therefore, a detector calibrated in a similar size to the chamber test as in this study is more reliable for assessment of ^220^Rn and ^222^Rn progeny exposure. At last, since the detector is sensitive to the exposure environmental condition, in the site survey, it is recommended to use the improved passive CR-39-based direct progeny detector after its calibration in a typical indoor environment.

## Figures and Tables

**Figure 1 ijerph-17-08569-f001:**
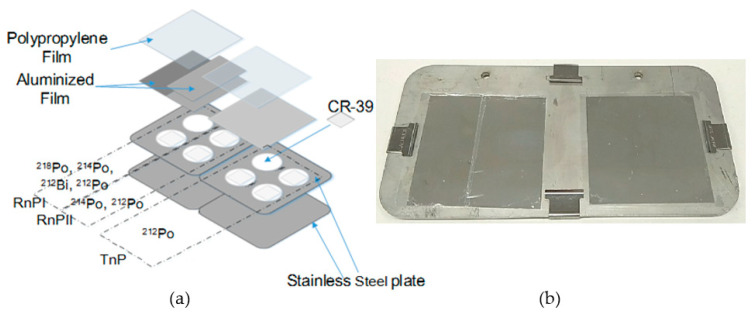
The passive CR-39-based direct ^222^Rn/^220^Rn progeny detector: (**a**) a schematic diagram of the progeny detector showing the detectable radionuclides of each channel and (**b**) a photo of the actual setup of the progeny detector.

**Figure 2 ijerph-17-08569-f002:**
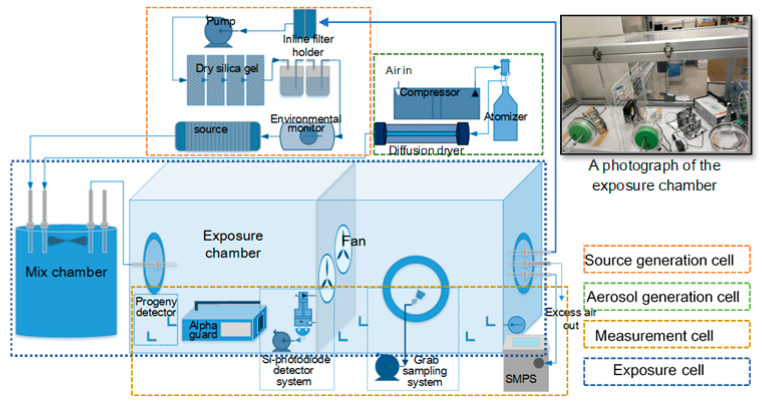
A schematic diagram of the chamber verification system.

**Figure 3 ijerph-17-08569-f003:**
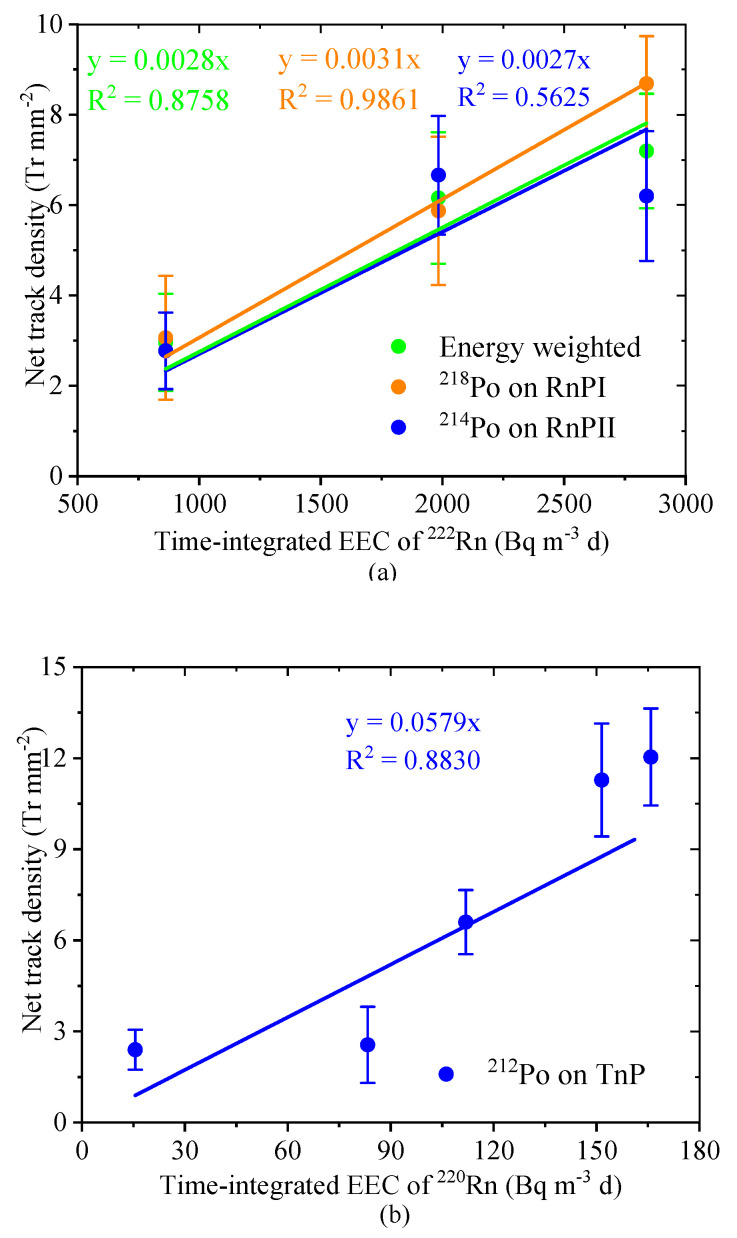
The net track density (NTD) as a function of time-integrated equilibrium equivalent concentration (EEC): (**a**) the NTD as a function of time-integrated EEC of ^222^Rn, where the NTD has excluded the contribution of other radionuclides on the CR-39, and (**b**) the NTD as a function of time-integrated EEC of ^220^Rn.

**Table 1 ijerph-17-08569-t001:** The design information of each detector channel.

Channel	Membrane Thickness (mg cm^−2^)	Nuclide	Energy of α-Particle (MeV)	Track Registration Efficiency, *η*	Deposited Atoms
TnP	7.10	^212^Po	8.785	0.063	^212^Pb, ^212^Bi
RnPII	5.05	^212^Po	8.785	0.212	^212^Pb, ^212^Bi
		^214^Po	7.687	0.182	^218^Po, ^214^Pb, ^214^Bi
RnPI	3.25	^212^Po and ^212^Bi	8.785 and 6.051	0.152	^212^Pb, ^212^Bi
		^214^Po	7.687	0.249	^218^Po, ^214^Pb, ^214^Bi
		^218^Po	6.003	0.135	^218^Po

**Table 2 ijerph-17-08569-t002:** The comparison results of the progeny detector and experiment in a ^220^Rn chamber verification system.

		Model	Experiment	Model	Experiment
Exposure time (h)		25.5		36.9	
The number concentration (N cm^−3^)		27,297		30,417	
GSD, σ_g_		1.506		1.613	
CMD (μm)		0.203		0.203	
Track density (tr mm^−2^)	Upward	3.9		2.9	
	Vertical	4.3		4.8	
Deposition velocity (m h^−1^)	Upward	0.185	0.143 ± 0.029	0.243	0.193 ± 0.067
	Vertical	0.169	0.158 ± 0.040	0.238	0.217 ± 0.045
	^212^Pb	0.185	——	0.219	——
	^212^Bi	0.132	——	0.183	——
Deposition flux (atoms cm^−2^ s^−1^)	Upward	0.081	0.081	0.055	0.069
	Vertical	0.090	0.090	0.042	0.042
EEC of ^220^Rn (Bq m^−3^)	Upward	237 ± 88	244 ± 108 ^1^	148 ± 38	155 ± 67 ^1^
	Vertical	263 ± 88		188 ± 50	

^1^ The value was calculated by the average equilibrium equivalent concentration (EEC) of ^220^Rn over the whole exposure period by grab sampling method.

**Table 3 ijerph-17-08569-t003:** The comparison results of the progeny detector and experiment in a ^222^Rn chamber verification system.

			Model	Experiment	Model	Experiment	Model	Experiment
Exposure time (h)			30		70		102	
Concentration of nuclei number (N cm^−3^)			34,528		33,833		33,278	
GSD, σ_g_			1.714		1.761		1.768	
CMD (μm)			0.233		0.200		0.191	
Track density (tr mm^−2^)	Upward	RnPI	6.3 ± 1.3		16.4 ± 2.5		14.9 ± 2.3	
		RnPII	4.5 ± 1.2		6.2 ± 1.6		4.9 ± 1.1	
	Vertical	RnPI	7.4 ± 1.5		13.6 ± 2.1		17.2 ± 2.7	
		RnPII	1.1 ± 0.8		7.1 ± 1.8		5.9 ± 1.3	
Deposition velocity (m h^−1^)	Upward		0.136	——	0.123	——	0.125	——
	Vertical		0.131	——	0.117	——	0.119	——
	^218^Po		0.290	——	0.248	——	0.230	——
	^214^Pb		0.155	——	0.135	——	0.112	——
	^214^Bi		0.101	——	0.091	——	0.092	——
Deposition flux (atoms cm^−2^ s^−1^)	^218^Po to RnPI	Upward	0.041	0.051 ± 0.010	0.022	0.048 ± 0.017	0.018	0.030 ± 0.016
		Vertical		0.043 ± 0.009		0.040 ± 0.014		0.035 ± 0.019
	^218^Po, ^214^Pb, ^214^Bi to RnPII	Upward	0.014	0.005 ± 0.004	0.011	0.014 ± 0.008	0.008	0.007 ± 0.006
		Vertical		0.023 ± 0.006		0.016 ± 0.009		0.009 ± 0.007
EEC of ^222^Rn (Bq m^−3^)	Upward		1214 ± 322	740 ± 27 ^1^	730 ±290	693 ±16 ^1^	758 ± 370	686 ± 31 ^1^
	Vertical		267 ± 187		603 ±312		630 ± 375	

^1^ The value was calculated by the average EEC of ^222^Rn over the whole exposure period by the continuous portable Si-photodiode detector.

**Table 4 ijerph-17-08569-t004:** Summary of the deposition velocities of ^222^Rn and ^220^Rn progenies in previous studies.

Reference	Deposition Velocity (m h^−1^)		Remark
	RnP Detector	TnP Detector	
[27]	8.64	19.08	Fan on, 26 m^3^ test facility
	3.24	2.16	Fan off, 26 m^3^ test facility
[7]		0.19 ± 0.04	Dwelling, the same structure of TnP detector as this study
[15]	0.132 ± 0.0036	0.075 ± 0.0072	22.5 m^3^ test room
	2.37 ± 0.785		0.5 m^3^ chamber
	0.117	0.072	Dwelling
[11]		0.828 ±0.072	150 L chamber, same TnP detector as this study
This study	0.125 ± 0.007	0.178 ± 0.045	3.24 m^3^ chamber

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
