# Peer review of "An Improved Passive CR-39-Based Direct ^222^Rn/^220^Rn Progeny Detector"

_ijerph, 2020, doi:10.3390/ijerph17228569_

Round 1
Reviewer 1 Report
Entire document: The term monitor is commonly used for electronic devices that provide a value
every time. I this case from my point of view you have built a detector or system. The terminology
used in the references for this kind of device is passive detector. Please change the term monitor
by detector.
Line 34: When yoy say "monitor chamber" are you refering yo the diffusion chamber of the passive
detectors of the monitor chamber of an AlphaGUARD for example. Hor can you control the air exchange
rate of those chambers?
Author Response
Dear Editor and Reviewer 1:
We highly appreciate your detailed and valuable comments on our manuscript. The suggestions and comments are quite helpful for us to modify this manuscript. Our responses are given in a point-by-point manner in the file of response to reviewers. Changes to the manuscript are shown by underline in the file of the revised manuscript with changes marked.
Point 1: The term monitor is commonly used for electronic devices that provide a value every time. I this case from my point of view you have built a detector or system. The terminology used in the references for this kind of device is passive detector. Please change the term monitor by detector.
Response 1: We revised the “progeny monitor” to the “progeny detector” throughout the manuscript. Please see lines 3, 16, 53, 63, 67, 75, 87-89, 227, 241, 243, 246, 270, 272, 317, 325-327, 330, 344.
Point 2: Line 34: When you say "monitor chamber" are you referring to the diffusion chamber of the passive detectors of the monitor chamber of an AlphaGUARD for example. or can you control the air exchange rate of those chambers?
Response 2: The monitor chamber here is referring to the diffusion chamber of the aforementioned passive detectors. The detectors make use of the different diffusion rates of the two chambers to distinguish 220Rn and 222Rn. The section has been revised to “In the two different diffusion rate chambers of these measuring devices, because there is a diffusion barrier against 220Rn based on its quite short half-lives, the low diffusion rate chamber detects mainly 222Rn, and the high diffusion rate chamber detects both 222Rn and 220Rn” to avoid confusion. Please see lines 34-37.
Reviewer 2 Report
The work entitled “An improved passive CR-39 based direct 222Rn/220Rn progeny monitor” regards a method based on a number of SSNTD CR39 covered by Mylar films of different thickness acting as absorbers. By this way 3 channels were created recording alpha particles of energy higher than specific values so, each channel was corresponded with specific radon/thoron progenies. Deposition velocity, flux and EEC for both radon and thoron progenies were experimentally derived and theoretical estimated based on a simple theoretical model using Jacobi’s room model for necessary parameters calculation. The theoretical and experimental results were found in good agreement. Consequently the authors presented typical and brief discussion of the results comparing them with values from the literature and studying the relation of NTD with time integration EEC over specific periods for 220Rn and 222Rn progenies.
The novelty of this work is not much higher than an average level and the applicability of the proposed method is restricted as it is depended on several environmental parameters and re-calibrations are required. However, the indirect measurement of 218Po and its involvement in EEC estimations is considered a significant output of the work. The work is suggested for publication in the “Int. J. Environ. Res. Public Health” and the authors are kindly advised to consider the following minor issues:
- The deposited on Mylar films radon progenies may emit alpha particles in a wide range of angles. Different angles result in different paths through the absorber so according to the emitting angle different portion of the initial energy of the alpha particle will be absorbed. Please clarify how the energy dependency on the angle of emission is regarded.
- The figure 2 where the experimental chamber is presented needs re-design in order to by easily readable and informative. Also the corresponding part of the manuscript should be improved. Also refer the volumetric activity concentrations of 222Rn and 220Rn produced by the sources (uranium rock and lantern mantles) and clarify if their ratio are in typical for indoor environment values.
- The authors referred p.2 50-53 “Even CR-39 is the most sensitive and popular detector for recording α-particles [14] and performs stable results, no direct both 222Rn and 220Rn progeny monitor using CR-39 was developed and applied, because its energy windows for recording α-particles is quite wide, and it is unable to distinguish alpha energy directly.” It is true that direct alpha particle spectroscopy has not yet achieved by means of CR39 however, related studies have suggested methods based on tracks geometrical characteristics and developed dedicated tools promoting the efforts for that aim. Should the authors consider for future works the exploitation of such tools in their studies?
Author Response
Dear Editor and Reviewer 2:
We highly appreciate your detailed and valuable comments on our manuscript. The suggestions and comments are quite helpful for us to modify this manuscript. Our responses are given in a point-by-point manner in the file of response to reviewers. Changes to the manuscript are shown by underline in the file of the revised manuscript with changes marked.
Point 1: The deposited on Mylar films radon progenies may emit alpha particles in a wide range of angles. Different angles result in different paths through the absorber so according to the emitting angle different portion of the initial energy of the alpha particle will be absorbed. Please clarify how the energy dependency on the angle of emission is regarded.
Response 1: In this manuscript, we introduced the parameter “η” to express the geometry efficiency of the detector. Otherwise, because the etching condition also affects the removing sickness of the CR-39, then affect the geometry efficiency, therefore in this manuscript, the “η” is defined as the track registration efficiency, which is the multiplier of the branching ratio of its progeny, geometry efficiency and etching efficiency. Please see Table 1. The corresponding section has been revised to “The track registration efficiency measured in this research is present in Table 1, which is corresponding to the etching condition using 6 N NaOH solutions at 60 oC for 24 h without stirring” to provide the additional measuring information of “η”. Please see lines 106-107.
Point 2: The figure 2 where the experimental chamber is presented needs re-design in order to by easily readable and informative. Also the corresponding part of the manuscript should be improved. Also refer the volumetric activity concentrations of 222Rn and 220Rn produced by the sources (uranium rock and lantern mantles) and clarify if their ratio are in typical for indoor environment values.
Response 2: The figure 2 had been revised and rearranged into 4 function cells. Otherwise, it should be noted that in this experiment, the chamber test only used the single source rather than the mixed 222Rn and 220Rn source in the exposure chamber. The section in the text has been revised to “The chamber verification system includes 4 function cells, as shown in Figure 2. (1)The source generation cell is a 222Rn gas and 220Rn gas generation system, which employ the natural uranium rock and commercially available lantern mantles as the 222Rn and 220Rn source, respectively to generate and import 222Rn and 220Rn to the mix chamber. The humidifier in this cell is responsible to maintain the stability of the inlet gas concentration. (2) The aerosol generation cell contains 3 components, a compressor, a constant output atomizer (TSI, Inc., Aerosol Generator Model 3076), and a diffusion dryer filled with silica gel. The 5000 μg mL-1 NaCl solution is employed to generate droplets with the number median diameter of 0.3 μm (GSD < 2.0), to mix with 222Rn gas or 220Rn gas in the mix chamber. (3) The exposure cell includes two chambers, mix chamber and exposure chamber. The volume of the mix chamber and exposure chamber is 150 L and 3.24 m3 (the dimension of 2.25 ×1.2 ×1.2 m3, L × W × H), respectively. The mix chamber provides a buffer to create more chances for the aerosol particles and gas to form the stable progeny flux to the exposure chamber. To maintain stable concentration in the exposure chamber, the experiment adopted a circulating connection that applies the outlet of the exposure chamber as the inlet of the source generation cell with an airflow rate of 3 L min-1. (4) The measurement cell is responsible to measure the progeny concentrations and environmental parameters in the verification experiment. The calibrated portable Si-photodiode detector with a sampling flow rate of 0.5 L min-1, and an EMD Millipore MF-Millipore AAWG02500 Mixed Cellulose Ester Filter Membrane were used to measure the EEC of 222Rn in real-time. The details of the Si-photodiode detector can be found elsewhere [23,24]. The periodic grab sampling method was carried out to replace the calibrated portable Si-photodiode detector to measure EEC of 220Rn”. Please see lines 197-217.
Figure 2. A schematic diagram of the chamber verification system.
Point 3:The authors referred p.2 50-53 “Even CR-39 is the most sensitive and popular detector for recording α-particles [14] and performs stable results, no direct both 222Rn and 220Rn progeny monitor using CR-39 was developed and applied, because its energy windows for recording α-particles is quite wide, and it is unable to distinguish alpha energy directly.” It is true that direct alpha particle spectroscopy has not yet achieved by means of CR39 however, related studies have suggested methods based on tracks geometrical characteristics and developed dedicated tools promoting the efforts for that aim. Should the authors consider for future works the exploitation of such tools in their studies?
Response 3: Thanks a lot for your suggestion. We will consider using such tools in future work to measure the 220Rn and 222Rn progenies precisely. Accordingly, the introduction section has revised to “Even CR-39 is the most sensitive and popular detector for recording α-particles [14] and performs stable results, no direct both 222Rn and 220Rn progeny detector using CR-39 was developed and applied, because its energy windows for recording α-particles is quite wide, and it is unable to distinguish alpha energy directly, although some researchers make great efforts to develop dedicated tools based on the track geometrical characteristics to measure alpha particles”, please see lines 52-56.
Reviewer 3 Report
Great job, i require more data about the radon Chamber (type of source, rate Bq/m3/h, flux of Air, etc).
Author Response
Dear Editor and Reviewer 3:
We highly appreciate your detailed and valuable comments on our manuscript. The suggestions and comments are quite helpful for us to modify this manuscript. Our responses are given in a point-by-point manner in the file of response to reviewers. Changes to the manuscript are shown by underline in the file of the revised manuscript with changes marked.
Point 1: I require more data about the radon Chamber (type of source, rate Bq/m3/h, flux of Air, etc).
Response 1: To express the verification experiment more clearly, we revised the figure 2 and the section in the text has been revised to “The chamber verification system includes 4 function cells, as shown in Figure 2. (1)The source generation cell is a 222Rn gas and 220Rn gas generation system, which employ the natural uranium rock and commercially available lantern mantles as the 222Rn and 220Rn source, respectively to generate and import 222Rn and 220Rn to the mix chamber. The humidifier in this cell is responsible to maintain the stability of the inlet gas concentration. (2) The aerosol generation cell contains 3 components, a compressor, a constant output atomizer (TSI, Inc., Aerosol Generator Model 3076), and a diffusion dryer filled with silica gel. The 5000 μg mL-1 NaCl solution is employed to generate droplets with the number median diameter of 0.3 μm (GSD < 2.0), to mix with 222Rn gas or 220Rn gas in the mix chamber. (3) The exposure cell includes two chambers, mix chamber and exposure chamber. The volume of the mix chamber and exposure chamber is 150 L and 3.24 m3 (the dimension of 2.25 ×1.2 ×1.2 m3, L × W × H), respectively. The mix chamber provides a buffer to create more chances for the aerosol particles and gas to form the stable progeny flux to the exposure chamber. To maintain stable concentration in the exposure chamber, the experiment adopted a circulating connection that applies the outlet of the exposure chamber as the inlet of the source generation cell with an airflow rate of 3 L min-1. (4) The measurement cell is responsible to measure the progeny concentrations and environmental parameters in the verification experiment. The calibrated portable Si-photodiode detector with a sampling flow rate of 0.5 L min-1, and an EMD Millipore MF-Millipore AAWG02500 Mixed Cellulose Ester Filter Membrane were used to measure the EEC of 222Rn in real-time. The details of the Si-photodiode detector can be found elsewhere [23,24]. The periodic grab sampling method was carried out to replace the calibrated portable Si-photodiode detector to measure EEC of 220Rn”. Please see lines 197-217.
Reviewer 4 Report
This manuscript presents an interesting and complete study based on an improved passive CR-39 design to measure EEC due to radon and thoron without knowing the equilibrium factor.
In general it is very well explained, concise and clear.
There are some comments detailed below:
- Please, confirm the units in Equation 2, just to be sure.
-
Line 131: superscripts in Vd does not look the same as in Eq.9
-
The "calibrated portable Si-photodiode detector" is mentioned several times in the text. Could you give some details about the calibration traceability?
-
It might be interesting to include a photograph of the actual setup of the CR-39 set, perhaps alongside Figure 1. And of the experimental setup (Figure 2).
-
Line 245: could you please indicate the main components of uncertainty that you have considered?
-
Table 2: Consider to express EEC (Tables 2 and 3) without decimal values.
-
Table 4: Bigu-1985 reference, is it correct in bold?
Author Response
Dear Editor and Reviewer 4:
We highly appreciate your detailed and valuable comments on our manuscript. The suggestions and comments are quite helpful for us to modify this manuscript. Our responses are given in a point-by-point manner in the file of response to reviewers. Changes to the manuscript are shown by underline in the file of the revised manuscript with changes marked.
Point1: Please, confirm the units in Equation 2, just to be sure.
Response 1: There were mistakes for the units in Equation 1 and 2. The units were revised, please see lines 101, 109-110.
Point 2: Line 131: superscripts in Vd does not look the same as in Eq.9
Response 2: This has been revised, please see line 136.
Point 3: The "calibrated portable Si-photodiode detector" is mentioned several times in the text. Could you give some details about the calibration traceability?
Response 3: To explain the calibration traceability of the calibrated portable Si-photodiode detector. The authors would like to refer to two related references in our group to let the readers could see more details of this detector easily. The corresponding section has been revised to “The details of the Si-photodiode detector can be found elsewhere [23,24]”. Please see lines 215-216 and 400-404.
Point 4: It might be interesting to include a photograph of the actual setup of the CR-39 set, perhaps alongside Figure 1. And of the experimental setup (Figure 2).
Response 4: Figure 1 and 2 had been updated by adding an actual photograph. Please see lines 86-89 (Figure 1) and 190-193 (Figure 2).
Point 5: Line 245: could you please indicate the main components of uncertainty that you have considered?
Response 5: We have added an explanation of the uncertainty in the estimation of the NTD. The corresponding section has been revised to “The calculation process of NTD itself has uncertainty, especially caused by the sensitivity of the CR-39 detector, read-out of the tracks, counting statistics, and assumed background of the CR-39 detectors [11], except for the exposure environment. Additionally, from eq (6) to eq (8), the calculation of the tracks registered by lower alpha-energy progeny should exclude the tracks registered by higher alpha- energy progeny on the channel. Therefore, the thinner the absorber filter is, the higher uncertainty of the measured deposition flux is present, due to the propagations of the uncertainty during the calculation of the NTD”. Please see lines 262-269.
Point 6: Table 2: Consider to express EEC (Tables 2 and 3) without decimal values.
Response 6: These sections have been revised, please see lines 270-273 (Table 2 and Table 3).
Point 7: Table 4: Bigu-1985 reference, is it correct in bold?
Response 7: The reference has been revised, please see line 286 (Table 4).